# The Value of Citizen Participation in Technology Assessment, Responsible Research and Innovation, and Sustainable Development

Nora Weinberger [1,*](ID), Silvia Woll [1], Christopher Conrad Maximillian Kyba [2](ID) and Nona Schulte-Römer [2]

1   Institute for Technology Assessment and Systems Analysis at the Karlsruhe Institute of Technology, 76133 Karlsruhe, Germany; silvia.woll@kit.edu
2   GFZ German Research Centre for Geosciences, 14473 Potsdam, Germany; kyba@gfz-potsdam.de (C.C.M.K.); nona.schulte-roemer@gfz-potsdam.de (N.S.-R.)
*   Correspondence: nora.weinberger@kit.edu; Tel.: +49-721-608-23972

**Abstract:** The participation of citizens in scientific research has a long tradition, and in some disciplines, especially medical research, it is even common practice. In Technology Assessment (TA), Responsible Research and Innovation (RRI), and Sustainable Development (SD), the participation of citizens can be of considerable value. In this paper, we explore this value for three concepts, based on the researcher's insights from three participatory research projects. The first project is the citizen science project TeQfor1, which was conducted with, for, and on the type 1 diabetes community, who do not feel adequately supported by the conventional health care system. In the second project, citizens with vision impairments participated in the technological development of an audio-tactile navigation tool in the TERRAIN project. The third project (Nachtlichter) dealt with light pollution. Based on the three projects presented, we show that citizen participation makes specific contributions to TA, RRI, and SD. We also investigate the specificity of citizen engagement and motivation by differentiating between existing and emerging involvement. In conclusion, we discuss the benefits that may be added by participatory approaches for the three concepts of TA, RRI, and SD.

**Keywords:** transdisciplinary research; citizen participation; citizen science; technology assessment; responsible research and innovation; sustainable development



## 1. Opening Up Science to Society

Transdisciplinary approaches, citizen participation, and stakeholder involvement are now integral parts of research and science funding, as well as innovation policies within EU policy. Compared to the disciplinary silos and ivory towers of the past, this "participatory turn" implies the increasing (political) role of deliberative initiatives and the need for accountability in science [1] (p. 235). As Jasanoff writes: "[t]he pressure for accountability manifests itself in many ways, of which the demand for greater transparency and participation is perhaps most prominent" (p. 236). This is also reflected in concepts such as Technology Assessment (TA), Responsible Research and Innovation (RRI) [2], and Sustainable Development (SD) [3]. Citizen engagement and public participation in the framing of research programs are just as (politically) demanded as the corresponding adaptation of research plans in order to achieve "socially shaped", sustainable, responsible and responsive socio-technical transitions [4] (p. 199). The underlying assumption is that participation is necessary in some cases (i) to solve (technological) conflicts and societal challenges, (ii) to ensure research takes into account the priorities and experiences of people affected by the research topic, (iii) to improve the accuracy and relevance of research (findings), (iv) to ensure people affected by the research issue benefit, (v) to shape technologies on the basis of societal values, and (vi) to increase the legitimacy and acceptance of policies (see inter alia [5]). However, citizen participation does not have

a single definition (e.g., [6,7]), and its *specific* values and contribution have not yet been empirically explored in relation to the overlapping but still distinct concepts of TA, RRI, and SD.

This (opinion) paper contributes to the closing of this research gap by exploring the contribution of citizen participation in light of the concepts of TA, RRI, and SD. Thereby, we draw on our research experiences in three participatory and citizen science projects (In this article, we use the terms citizen and citizen scientists to refer to non-paid project members. We do not imply national citizenship [8]). Our conceptual research question is: what is the added value of citizen participation in the three projects with regard to TA, RRI, and SD? This question is closely related to more practical aspects of individual "affectedness", personal concern, and engagement (e.g., why do people participate in research projects in the first place?). To explore these related questions, we start with a description of the concepts of TA, RRI, and SD and their commonalities and differences (in relation to citizen participation). Based on this comparison and our reflections on three participatory research projects, we then outline from the direct and reflected experience as researchers how citizens contributed to the projects and consequently illustrate that the claimed benefits of participation implied in the concepts of TA, RRI, and SD are valid. We elaborate how participation generates an added value for science, as well as for the citizen scientist, and reflect on its value in light of the respective concepts. We also discuss different degrees of participation, which can range from information-only events with a "participatory touch" [9] (p. 13) to collaboration as well as co-creation processes and co-design projects with citizens as co-researchers (e.g., [10,11]) widely discussed as "Citizen Science" (CS) and more recently Citizen Management, "where citizens, adequately trained in problem solving" [12] (p. 1).

The contrasting comparison of citizen participation in the three projects allows us to picture a variety of ways and degrees in which people can participate in and profit from reflexive and responsible technology development and sustainability-oriented research and innovation. Beyond that, we discuss factors that might affect people's engagement and motivation to participate in research, and variations in the "specificity of involvement" (participation of an interested public vs. patient participation). Finally, we discuss the specific values that citizen participation approaches, and in particular, decision-making power in the hands of citizens, can add to the three fields of TA, RRI, and SD.

## 2. Citizen Participation in TA, RRI, and SD

Although they are different concepts, TA, RRI, and SD share the core idea that knowledge plurality and the transdisciplinary engagement of citizens and other stakeholders lead to better scientific and technological solutions. It has been argued that the so-called "Grand Challenges" [13,14], such as climate change, nutrition, access to drinking water, and ageing societies, can only be addressed by moving beyond disciplinary research and technology development [15]. Nevertheless, the concepts of TA, RRI, and SD are related to different and specific challenges in sociotechnical transition and governance processes.

TA represents "the idea of designing, controlling and regulating technology with a view to social and ecological consequences" [16] (p. 11) [translation by the authors]. Integrating a variety of stakeholders into TA can help anticipate the potential impacts of technology on society and the environment and increase public acceptance. Technology development is addressed in "constructive TA", a toolset for bringing "[feedback] of TA activities into the actual construction of technology" [17] (p. 252). Constructive TA (cTA) uses "the social shaping of technology and incremental decision-making and attempts to improve the social robustness of decisions about technology" and "is best understood as an umbrella term encompassing a range of approaches to technology assessment underpinned by shared concerns about the emergence and control of risky or controversial technology" [18] (p. 434) (see also [17,19–21]). Methods for fostering participation are key to this constructive social shaping of technology. Participatory TA (pTA), on the other hand, involves "diverse social actors from academia, business, law, education, etc.". It calls "first

and foremost [ . . . ] [for] the inclusion of citizens" [22] (pp. 1–2) in its procedures, with the aim of democratizing science, research, and technology policy [23], in order to create social awareness for and to mitigate the unintended side effects of technology innovation and development.

The call for responsible research and innovation has been articulated especially in research policy contexts [16]. It implies societal participation at the early stage of envisioning research questions, technologies, and futures. As Sovacool et al. [24] (pp. 12–13) outline, RRI "is not only about understanding specific technologies in society but also about reshaping the whole research and innovation process, even before applications are considered". RRI enables, for instance, a responsive adaptation of research questions and innovation paths and the deliberative identification of potential (normative) ideals and moral controversies (e.g., [25,26]). Thus, "[r]esponsibility of research is about pursuing the aims of society ensured by constant exchange with relevant societal actors" [5] (p. 66). This offers the hope of using RRI to base technology and innovation on values that are socially regarded as positive.

Thus, as with TA, it is about "social shaping", but in contrast to TA, the focus in RRI is no longer on the question of which particular technology is critically evaluated or which individual technology is socially accepted [27]. Instead, RRI aims to rethink scientific knowledge production, technology, and innovation, and to answer the fundamental questions of which innovations are needed and which research should be promoted in a social discourse on values. For example, citizens can discuss what kind of future they want to live in and what values are important to them. "However, individual technologies should not be subjected to an ethical evaluation [ . . . ]. Rather, innovation as a whole should be subject to the primacy of ethics [ . . . ]" [translation by the authors] [16] (p. 12). Normative reference points here are, for example, sustainability, in addition to social desirability and acceptability.

Sustainable development overlaps with RRI, as both processes are grounded in scientific evidence production. Moreover, sustainability is often associated with the idea of responsibility towards future generations. Ladikas et al. [5] even argue that responsibility and sustainability are conceptually close and highly interchangeable. As with RRI, a normative and value-based guiding principle is the basis. However, while "sustainability" describes the *normative goal of the process*, the concept of SD draws attention to *the process of transforming* socio-ecological-technical systems. Understood as a process, SD highlights the need for transformative system change, and explores the transferability and upscaling of local solutions for global problems [28,29]. Scientific evidence is indispensable for defining "sustainability" and sustainable development goals (SDGs) (The 17 Sustainable Development Goals (SDGs) are political targets set by the United Nations to ensure sustainable development at the economic, social and environmental levels worldwide.). So is citizen participation, because these transformation processes towards sustainability are complex and affect society as a whole, so goal-conflicts and controversies are inevitable. This implies the integration of knowledge from different scientific disciplines as well as professional, practical, and citizen knowledge [30]. Because of this need for knowledge integration and openness to newness, SD is often practiced in experimental participatory formats such as urban laboratories, real-world experiments, and living labs [31–33]. "The involvement of citizens in scientific research and/or knowledge production, can complement and ultimately improve the SDG reporting process" [34] (p. 922) and improve the evidence-base for SD policies [35] (p. 373). It can also contribute to raising awareness for societal needs and environmental issues among both citizens and policy makers [36,37]. Citizens can thus challenge, contribute to, and shape the scientific knowledge that builds the evidence base for SD.

In conclusion, RRI, TA, and SD can be seen as related concepts, which align research with societal needs with different scopes. While TA is more about "understanding specific technologies in society", RRI as a normative concept addresses "the entire research and innovation process, even before applications are considered" and SD "takes a whole

sociotechnical systems approach to innovation policy, starting from ongoing developments in society that can be, through research and development, scaled to the systems level." [24] (pp. 12–13).

### 3. Citizen Participation in Practice

Against this backdrop, we present in the following three different projects in which citizens were engaged and empowered in research and innovation, and describe their specific participatory format and approaches. From our perspective as lead scientists in these projects, we outline personal involvements that drive citizen engagement, ranging from disease condition to environmental activism, and discuss how the participation provides value for the research or the participant from the authors' perspective. We also depict the different roles of citizens, and how these roles relate to their specialist knowledge (e.g., as patients, technology users, or co-researchers). The participatory research and citizen science projects we describe all relate to TA, RRI, and SD, so that our analysis does not aim at a one-to-one assignment of projects and concepts but focuses on similarities and differences.

### 3.1. Impacts of Technical Systems on the Personal Quality of Life of People with Type 1 Diabetes (TeQfor1)

TeQfor1 is an ongoing CS project conducted with, for, and on type 1 diabetes (T1D) community from German-speaking countries, who do not feel adequately supported by the conventional health care system [38]. T1D is a chronic condition where the pancreas no longer produces insulin. It is therefore managed by calculating requirements and administering insulin externally. Achieving the targeted blood glucose levels is extremely difficult, and requires lifelong constant self-control to avoid the potential for serious acute and long-term consequences. For some years, research has been conducted into developing artificial pancreas systems (APS), which enable automated insulin delivery according to blood glucose levels. The first commercial APS just recently hit the European market in 2018 [39]. Such systems make it much easier for people with T1D to deal with their disease. However, current APS versions are still far from actually replacing pancreatic function, due to technical and pharmaceutical limitations, but also due to safety restrictions preventing the use of the systems in a way that would actually be technically possible. In addition, these systems are not available for all persons with T1D, and have so far been developed mainly without their involvement.

In response, people with T1D and their relatives have formed a community under the hashtag #WeAreNotWaiting, expressing their conviction that they can develop better solutions for their specific requirements than those offered by the conventional health care system to date. The community innovatively develops their own systems based on commercial technologies and makes them freely available. These developments now enable an algorithm-based delivery of insulin doses adapted to continuously measure tissue glucose values, in order to keep blood glucose levels in a safe range. They are far more effective than conventional systems, and available free of cost. The user acceptance of the technology and the social shaping on the lifeworld perspective, the needs and values of the user (e.g., as aims of TA and RRI) can therefore already be assumed as given.

However, a systematic (scientific) evaluation of their effectiveness, e.g., with regard to improving the glycemic control and the quality of life is lacking. An evaluation of such systems by citizens living with T1D or caring for children with T1D is therefore of high relevance, especially since the people affected by T1D constantly have to deal with the disease and can bring this expertise to the development process.

TeQfor1 provides the users of such systems with a participatory approach that enables them to make sound and valid assessments of their DIY technologies, focusing on their own criteria. Citizens using any type of DIY APS for themselves or their children investigate how this use affects their blood glucose levels and their quality of life. The criteria or factors for assessing the quality of life are first defined by the citizens themselves. This

gives participants the opportunity to generate data for scientifically sound studies on an important topic that has so far been largely ignored by academic research.

In TeQfor1, the level of participation is very high, as the citizens are in control of every task of the project (except for the project proposal), and are highly encouraged to take leading positions. The academic scientists should only take on advisory positions.

### 3.2. Autonomous Mobility for Blind and Visually Impaired People in the Urban Space through Audio-Tactile Navigation (TERRAIN)

The project TERRAIN [40] aimed at generating "enabling structures" for blind and visually impaired people for orientation in urban spaces [41]. To this end, the project team developed a support and guidance system for orientation and navigation to strengthen the people's free movement, independence, and possibilities for interaction and thus their social inclusion. At the same time, this reduces the orientation risks, especially regarding dangerous barriers (e.g., road traffic or obstacles in chest or head height). Modern methods of image processing are used to "inform" the users acoustically, haptically, or tactilely in real time about their current environment. In this way, the blind and visually impaired people's areas of mobility and movement are extended.

Blind and visually impaired people and other non-academic stakeholders (e.g., mobility trainers) were involved in the development of this assistance system to ensure that the responsive and adaptive development process was based on user needs and everyday-life experiences (due to the focus of the paper, the project is only presented in an abbreviated form with reference to citizen participation. For example, the advisory board and the involvement of other stakeholders are not discussed. See, among others, [41]). In practical terms, a requirement analysis and two field test phases were carried out on a partnership level in a kind of pTA/cTA module, through which the development directions were continuously adapted. The project also focused on aligning the potential (un)intended effects of innovation outcomes (e.g., a use of cameras in public spaces for orientation) with important societal perspectives, values, needs, normative ideals and moral controversies. The aim was to evaluate the degree to which the technical innovation is responsible (in the sense of RRI and TA). Likewise, based on narrative descriptions of the social context of technology use and technological visions, citizens as an interested public (the random sample involved a citizen with sight impairment; see [41] for acquisition of participants) evaluated the potential for changes in the social fabric. This allowed for the fullest possible understanding of what future technology should be promoted in the specific context according to the citizens.

This participatory process also allowed non-visually impaired participants to experience different forms of visual impairment, with the help of simulation glasses provided by the German Association for the Blind and Visually Impaired. Some participants also chose to have a second person guide them through the workshop room in order to directly experience the feeling of "being dependent on someone" that is often expressed by those with visual impairment. These exercises seemed to result in the participants in the second phase of the module becoming more sensitized to the daily challenges of people with visual impairment, and subsequently also arguing more extensively from the perspective of those with visual impairments.

Furthermore, the ethical, legal, and social implications of the innovative assistance system were considered by an inter- and transdisciplinary panel of disciplinary and life-world experts (here: citizens with vision impairments) throughout the entire technology development process.

### 3.3. Citizen Science for Sustainable Lighting (Nachtlichter)

The Nachtlichter (night lights) citizen science project [42] uses a participatory research approach to quantify artificial light emissions, an increasingly recognized form of environmental change often referred to as light pollution [43,44]. In the Nachtlichter project, participants produce scientific data about how artificial light is actually applied in practice, using a co-designed web application. Local teams have been formed to count light sources

in several (primarily German) cities, thereby creating the first large-scale (~2 km$^2$ each) local inventories of (all) light sources in urban public spaces (existing lighting inventories are nearly entirely focused on street lighting, cf. [45]). The project develops the lighting expertise of lay participants through training activities that were co-designed by a core team of academic and citizen scientists. The online links to the Nachtlicher App and training tutorial is available on the project website [42].

Citizen scientists are involved in nearly every aspect of the Nachtlicher project, from the application for funds and app co-design to campaign planning and data collection. In the app co-design process, the transdisciplinary team created and tested a category system for counting all types of different light sources (e.g., streetlights, facade and shop lighting, traffic lights, illuminated windows) and information about their key characteristics (size, color, and shielding). This process involved discussions about what kind of data should be generated, and for what purpose (in the sense of RRI and TA). The main scientific goal remained the collection of geographical data on night light numbers and types, to better understand ("ground truth") satellite imagery. However, participants brought in and developed additional ideas about what data should be produced and how they could be used. Some aim to collect place-specific data to share with local decision-makers and politicians, with the hope of reducing light pollution in their city. Others expect that the data collection campaigns will raise public awareness for unsustainable lighting. One local measurement campaign team hopes to engage younger people in their dark-sky protection activities. A near-universal reaction by project participants has been that the experience changed the way they view outdoor scenes after dark, and they are now much more cognizant and critical of both public and private lighting infrastructure in their communities. Using the app for counting lights can thus be understood as not only a scientific contribution, but also as political "material participation" in SD [46].

The Nachtlicher project contributes to sustainable development in at least four ways. First, the app provides new data on the number and properties of public and private lights in public spaces. In the face of the ongoing LED transition towards what is meant to be more sustainable lighting [47], this information can offer important insights for best practice and the governance of artificial light at night in the future. Second, taking part in the project appears to raise citizens' awareness for the negative side effects of artificial light, and the difference between sustainable and problematic lighting installations. Third, Nachtlicher indirectly supports the creation of local networks that can advocate for light pollution reduction, as the project is centered on a small number of local campaigns with intensive community participation. Finally, the formation of transdisciplinary networks can encourage the up scaling of the approach, and the translation of the scientific data and the research results into sustainable lighting practices. To this end, the newly designed app and collected data will also be openly and freely available after the project, allowing future scientific analyses and additional data collection campaigns around the world.

## 4. Discussion: Different Voices

When contrasting the three citizen participation projects described above we see interesting similarities and differences that are closely related to our research questions. First, the projects differed with regard to their levels of participation in different phases of the research process (Table 1). Second, we observed varying degrees of personal involvement in the projects. Critical reflection on these differences allows us to address our practical questions regarding individual "affectedness" and personal concerns as a motivation to engage in research projects, as well as our conceptual question regarding the added value of citizen participation in the contexts of TA, RRI, and SD.

**Table 1.** Citizen participation in different project phases in TeQfor1, TERRAIN, and Nachtlichter.

| | TeQfor1 | TERRAIN | Nachtlichter |
|---|---|---|---|
| Consulting on a research topic | From the beginning of the project, a group of seven citizen scientists were included in all phases and decisions of the project, and also provided advice about the direction the project should take. | The idea for the technology development came from a brainstorming session between an industrial project partner and a visually impaired employee of a research institute that was later part of the later consortium. | Discussion with citizen scientists during the initial event fed into the research design. Citizens' interests in light pollution mitigation shaped their engagement, the app development, and the data collection process. |
| Developing (research) questions | The academic research team offered research questions to the citizen scientists. The final research questions were determined together with the group of seven citizen scientists by meeting several times and following their research interest. | In the run-up to the workshop, the interested citizens named relevant discussion topics from their point of view in the course of the written invitation. These topics were then collected and integrated into the preparation of the workshop content. | Citizen scientists were encouraged to raise and explore their own (research) questions. In response to their demands, the app includes a comment function where citizens can document their personal observation of light sources in specific public spaces. |
| Developing methodology and research tools | The academic research team offered options for the methodology and research tools to the citizen scientists. The final methodology and research tools were determined together with all citizen scientists in online meetings and in exchanges via an online platform. One final research tool was determined by the citizen scientists using a poll. | The academic research team selected the methodology and research tools. | The data generation methodology of the "Nachtlichter" app was co-designed with citizen scientists. This included the development of an easily comprehensible system for classifying light sources and their characteristics. |
| Data collection | (1) Questionnaire for aspects of quality of life (pending); (2) narratives of several everyday situations via two workshops and an online platform (collection took place, writing of narratives pending); (3) collection of (digital) blood glucose data via the infrastructure of a different project (pending). | Needs and requirement analysis and field tests with blind and visually impaired participants, non-impaired participant deliberative module. | Citizen scientists are the principle data collectors and count lights independently in public spaces using the Nachtlichter app. They also impact data quality, as they are trained and gain experience in using the app. |
| Analysis and interpretation of the findings | Pending; will be completed by all citizen and academic scientists together. | Citizens commented on and supplemented the workshop results. A series of in-depth interviews were conducted on individual topics (including visibility and funding of technology, privacy and data use, exclusion/inclusion) that citizens rated as particularly relevant. In addition, there were several adaptive and reflective loops in technology development with people with visual impairments. | App data will be used for the ground truthing of satellite imagery. Local data collectors will offer site-specific information. Citizens are invited to use the data generated in the project for their own interpretations, analyses, and purposes (e.g., dark sky certification). |

**Table 1.** *Cont.*

| | TeQfor1 | TERRAIN | Nachtlichter |
|---|---|---|---|
| Project management | Academic researchers and citizen scientists equally. | Project management was in the hands of the academic scientists. | Academic researchers organize meetings and legal matters, local campaign organizers run own their events. |
| Writing up and reporting | Pending; will be completed by the academic researchers but with support and feedback from the citizen scientists. | Academic researchers will take responsibility and the lead. | Academic researchers will take responsibility and the lead, with participants as co-authors. |
| Dissemination of project results | Pending; the results will be disseminated by the academic researchers and by the citizen scientists each via their respective channels/structures (as the citizen scientists are free and encouraged to work with the results as the academic researchers do), but also in a common effort. | The results were disseminated by the academic researchers and discussed in relevant communities. | Open access publication of data and project results. Citizen scientists are involved in the public presentation of project results in academic conferences and public forums. They will also help disseminate results via their networks (dark sky and environmental protection groups). |

### 4.1. Levels of Citizen Participation

In participation research, levels of participation are often evaluated with regard to citizen engagement in different project stages. High levels of participation are thereby associated with early and/or continuous engagement (see, e.g., [9,11]). Based on this conceptualization, the three projects showed varying levels of participation.

In the TERRAIN project (see Table 1), citizens with vision impairments were involved in the field test phases, several advisory board meetings, and interviews. Citizens without visual impairments participated in a workshop and were involved in several "consultation" modules, e.g., advisory board sessions. Furthermore, the citizens were invited to define the research questions of the workshop, but the academic researcher took the lead in most scientific decisions. In TeQfor1, citizens with T1D were involved in all phases of the project from the beginning, with full decision-making power. This, in Arnstein's view, this high level of "citizen control" [9,11] was also the case for the Nachtlichter project. In both TeQfor1 and Nachtlichter, the citizens were involved from the beginning in consulting on the research topic over the development of the research questions and methodology to the dissemination of the results. It can therefore be noted that citizens were involved in the three projects as co-designers (app development in Nachtlichter and technology development TERRAIN) and as co-researchers (Nachtlichter and TeQfor1).

A high degree of participation in research and innovation, e.g., in approaches such as Citizen Science, is very popular right now. Nonetheless, even projects such as TERRAIN, in which the level of participation was lower compared to the other two projects following Arnstein [11], led to socially shaped and accepted technological innovations, which met the expectations of the TA and followed the principles of RRI. For instance, through extensive consultation in partnership by citizens with and without visual impairments, a responsible technological solution could be innovated in a discourse of social values. For example, citizens with and without visual impairments have debated and have come to a common understanding regarding societal needs and values such as inclusion (acceptance of social diversity, compassion, equality and treatment, community), independence, and autonomy that are crucial to the way they want to live (together) in the future.

However, based on our research experiences and the comparison of the three projects, we have found that project success does not depend on high levels of participation. Instead, the appropriate level of participation always depends on the research question and the goal of the innovation. Thus, in the TeQfor1 project, a high degree of decision-making power in the hands of citizens was paramount for appropriately designing technological innovations that met the everyday needs of citizens with T1D. Especially in the context of "DIY technologies" for "living with T1D", we take the view that people with T1D are clearly the best assessors, as they use the technology or have even developed it themselves. In addition, as patients, they are the experts on the daily challenges of T1D. In conclusion, a high level of participation seems advisable for TA where people are individually (health) affected, in order to achieve the constructive social shaping of the technology by connecting it to the lifeworld in all project phases, e.g., to identify the right research questions and to evaluate the technology (vision/s) (see Section 4.2 in particular). For transformation processes towards sustainability (SD), citizen participation is a salient way to integrate personal concerns to perform environmental research responsibly, and to deal better with conflicting goals and the complexity of transformation challenges. For example, the Nachtlichter project contributes to the development of an evidence-based and consensual definition of sustainable lightning. It appears that the participants' experiences and deepened expertise in outdoor lightning helps them to articulate and defend their vision for sustainable lighting, for example in discussions with decision-makers and in public discussions. Moreover, citizen science projects such as Nachtlichter can offer approaches for upscaling data collection for evidence-based sustainable transition efforts that could otherwise not be obtained.

In the next section, we use an inductive comparison of the three projects to illustrate the personal involvement of citizens, and discuss the specificity of citizens' knowledge.

### 4.2. Existing and Emerging Personal Involvement

In our participatory projects, personal involvement and citizen expertise mutually reinforced each other. In all three projects, citizens appeared to be more motivated and more actively engaged when they could contribute their specific knowledge and expertise. In the following, we describe the differences in this process and thereby distinguish between what we call existing and emerging involvement.

*Existing involvement* was strong in the projects TeQfor1 and TERRAIN, where citizens with T1D or visual impairment participated. This involvement coincides with a specific knowledge of what it means and requires to master everyday routines as an ill or disabled person. Based on their everyday experiences with the impairment and disease, participants contributed their specialized knowledge to the research and technology development in the two projects. People with visual impairment or T1D and their close relatives are experts in handling the respective technologies that alleviate their health conditions, and the impacts these have on their lives. They know which technologies and techniques are available to them and are well aware of their shortcomings. In the case of TeQfor1, all participants had a direct lived experience of the disease. Moreover, the open source technologies were the result of personal involvement and DIY technology design that further facilitated co-development within an already existing community. Some members of this community seemed to be excited to conduct research with academics as a partnership of equals, and to eventually produce scientific evidence about the positive impact and safety of the technologies. In the TERRAIN project, visually impaired people contributed to the requirements and needs analyses. They also participated in field tests where they used cameras in the streets and together with the scientific team explored whether the innovation could be integrated into their lifeworld and would correspond with existing norms and values.

*Emerging involvement* was observable in situations where citizens gained knowledge in the course of the projects, and thereby intensified their engagement. We observed this emerging involvement in the TERRAIN as well as in the Nachtlichter project in participants that were not personally or physically affected by the issues at stake, but interested in the project topics or scientific research. In the TERRAIN process, interest in participatory citizen workshops was so high that participants had to be selected through a lottery. These representatives of an interested public were asked to assess whether the use of an innovative camera application for visually impaired people was acceptable in public spaces and discussed potential side effects, for example. Since these workshop participants were mostly not personally affected by visual impairment, it also raised awareness of the difficulties that visually impaired people encounter in public spaces. To this end, participants moved around with simulation glasses and tried "accompanied walking through unknown space" while using these glasses. We could observe how this experience brought the lifeworld's of healthy and impaired participants closer together, and triggered normative discussions on how technological innovations could enhance social inclusion and mobility. It also appeared to have an effect on the citizens' personal involvement. In the later workshop discussions, taking the perspective of people with visual impairment was a key issue [41]. For example, the researchers in the TERRAIN project discussed with citizens without visual impairment how the workshop exercises allowed them to develop a sense of "being dependent on someone", and to experience the consequences of different visual impairments on the perception of the surroundings.

In Nachtlichter, most citizens involved in the co-design of the Nachtlichter app were motivated by a desire for healthy ecosystems or a starry night sky, but in contrast to the projects TeQfor1 and TERRAIN, they were generally not physically affected by outdoor lighting. When asked why they engaged in the citizen science project, many participants said they were concerned about the negative side effects of artificial light at night on

wildlife, humans, and entire ecosystems (cf. [48]). Many participants also mentioned that they loved stargazing, and some were even dark-sky activists. During the co-design process, we could observe how their personal involvement increased as their contributions to the app design process really made a difference. The knowledge and experiences they brought to the table were diverse. Some were experienced dark-sky activists and had expertise in sustainable lighting. One participant contributed to the project as a geodata expert, another produced a video. A participant who felt insecure in public spaces at night shared her experience so that we could design the project in a more inclusive and user-friendly way. The participants also developed a new awareness for lighting infrastructures in their nighttime surroundings. Several reported that before the project, they had never paid attention to the great variety of light sources in their direct neighborhood. Some even became ambassadors of Nachtlichter and presented the project and its goals during scientific conferences and public events. They repeatedly noted that they had thereby gained new (presentation) skills and confidence.

In all three projects, citizen engagement was noticeably spurred through collaboration framed as a partnership of equals, mutual respect, and learning in the transdisciplinary team. Regarding the value of participation from the citizen's perspective, it seems that the scope for engaging people not only depends on the topic and existing engagement, but also on the process of participation and transdisciplinary exchange [49]. In all three projects, participants seemed most involved in the projects where they could actively contribute with their relevant knowledge to TA, RRI, or data collection related to SD. Whether they developed this expertise during the project or joined the projects with prior knowledge and experience was not particularly decisive.

## 5. Conclusions: The Value of Citizen Participation

The added value for science through the involvement of citizens is well known [50–52]. The three projects presented in the paper add to this, by highlighting the importance of reducing the separation of science and society, and revealing the advantages of involving citizens in research and innovation. This is in line with an extensive literature describing the benefits of inviting heterogeneous actors with different knowledge and expertise and discussing the challenges of facilitating such processes (see, among others, [1,4,15,30,49,53]). This supports the arguments of Epstein [54] and Callon [55]. In this context, Epstein describes it as erroneous to think of the role of citizens as merely passive [54]. Instead, in his opinion, their contributions are an important resource in the process of knowledge creation. Besides, Callon has pointed out that the separation of science and society undermines trust in science [55]. In our view, this has also been evident during the COVID-19 pandemic, as if under a focal glass. This separation needs to be addressed and, if possible, resolved, by involving citizens more in the aspects of research (and policy) [55].

Taken together, the participatory research projects in this paper show that involving citizens in science provides evidence that could not otherwise be achieved, and shapes the direction of research by generating new research questions. This is highly relevant with regard to RRI, because citizens should be the long-term beneficiaries of public research finding. As we can tell from our own project experience, citizens challenge scientists to evaluate the relevance of research questions and approaches, particularly where it pertains to their lived experience [5]. Focusing on the concepts of TA and SD, we conclude from our projects that the added value of participation is particularly high where scientific evidence and technology developments affect people in their everyday lives [30,55] (whether because they are affected by a health impairment or limitation, patients, or concerned about social-technical or socio-ecological transformations in their lived environment). In such contexts, stakeholder participation and citizen science can provide important insights for sustainability-oriented scientific evidence production (SD) and a more inclusive social shaping of technologies and the applications of research findings (TA) [4].

Focusing on the practical aspects of participation for RRI, TA, and SD, our comparative project reflections suggest that the "right" level of participation is highly research topic-

specific or project-specific. There are cases where citizens start off research and undertake science or technology development before any academic project even exists. In TeQfor1, for instance, citizens contributed to the technology development as well as the TA; they just did not undertake the assessment with standard academic techniques. TeQfor1 could build on these approaches and follow them up with scientific advice on methods and quality criteria. In some cases, researchers rely on engagement in order to collect unique data they could not obtain otherwise. In the participatory research project TERRAIN, matching the developed technology to the users' needs could only be assured by involving them on a co-design level, therefore guaranteeing a broadly accepted outcome of the technology development. The same applies to the data collection in the Nachtlichter citizen science project, where citizen participation allows research on (spatial) scales that would not be possible without citizen engagement, for example in the lighting inventories of Nachtlichter.

However, to ensure that the contribution of citizens in scientific projects serves both sides—science and citizens—and that participatory research develops constructively, we consider it highly relevant to systematically evaluate participatory projects. In these, both the methodological approaches and the results should be assessed also with regard to the resources available for science communication and the facilitation of participation processes and formats. Such an evaluation should be carried out independently of the level of participation of the respective project, and reflect both the perspective of the (academic) scientists and the citizen researchers. Our experience with participatory research projects and citizen science clearly indicates that the added value of participation is a mutual enrichment: it is not only of value for academic researchers, but also for the citizens themselves (see, e.g., [12]. In our experience from the projects, citizens understand how they can contribute and how they can interact. They gain insights into how science works and into the methods and approaches used to conduct science. In particular, they learn about scientific uncertainty—the future they are discussing may not take place. They also have the experience of moving out of their comfort zone and taking new roles. Thus, our observations allow us to assume that project participants gained new knowledge, had experiences of self-efficacy, and felt they contributed to the common good. For instance, in the TERRAIN projects, citizens were pleased to gain more awareness about the perspective of blind people and visually impaired people, and to speak out on their behalf. In the Nachtlichter project, citizen scientists enjoy deepening their knowledge in their subjects of interest and concern such as light pollution and dark sky protection. However, despite these mutual benefits, the vast majority of participatory projects lack such a systematic evaluation. We therefore propose that symmetric project evaluations could be a constructive way to determine the value of participation in relation to the level of participation, and provide more insights in the concept and value of participatory research.

Even without such a systematic evaluation, the results and our experiences as academic scientists allow us to delineate the added value of citizen participation in TA, RRI, and SD. In TA, participation challenges technology developers to find more inclusive and acceptable solutions, as the citizens are already involved in the needs assessment, sometimes before there is even an academic or industrial idea for a technological innovation. They contribute by reflecting upon (technical) solutions for the challenges they face in their daily lives. In the case of RRI, alignment with values and moral concepts leads to a societal consensus on how to conduct research and innovation in a responsible way, which in turn leads to responsible technology development. The question here, however, is whether it is sufficient to undertake this at various points in time during a project, or whether continuous involvement (in citizen control) is required, as Arnstein [11] calls for. In the context of systematic transformations towards sustainability, citizen participation fulfils an important legitimizing function. Transformative projects such as sustainable lightning, energy transitions, climate protection measures, and mobility change can only succeed on a macrosocial scale if citizens participate and approve of the transition processes. After all, citizens will have to adapt to and help promote new practices in their everyday lives—by using public transport instead of private cars or by consuming energy differently

or less intensively. In short, solving societal challenges requires societal engagement. Only together can we decide how we want to live in the future. In our experience, working with citizens promotes critical reflection on the way we study science.

**Author Contributions:** Conceptualization, N.W., S.W., C.C.M.K. and N.S.-R.; methodology, N.W., S.W., C.C.M.K. and N.S.-R.; investigation, N.W., S.W., C.C.M.K. and N.S.-R.; writing—original draft preparation, N.W., N.S.-R.; S.W. and C.C.M.K.; writing—review and editing, N.W., S.W., C.C.M.K. and N.S.-R.; project administration, N.W., S.W. and C.C.M.K.; funding acquisition, N.W., S.W. and C.C.M.K. All authors have read and agreed to the published version of the manuscript.

**Funding:** This research was funded by the Federal Ministry of Education and Research (TERRAIN, 16SV7611), and the Helmholtz Association Initiative and Networking Fund (TeQfor1, CS-005, and Nachtlichter, CS-003).

**Institutional Review Board Statement:** The projects TeQfor1 and Terrain were conducted according to the guidelines of the Declaration of Helsinki, and approved by the Ethics Committee of the Karlsruhe Institute of Technology. Institutional review was waived for Nachtlichter, because it does not involve research on human subjects.

**Informed Consent Statement:** Informed consent was obtained from all subjects involved in the projects.

**Data Availability Statement:** No new data were created or analyzed in this study. Data sharing is not applicable to this article.

**Acknowledgments:** The authors wish to thank the entire consortia of the projects TeQfor1, TERRAIN, and Nachtlichter in which the results presented here were gathered. Above all, we would like to thank the citizens, patients, and other participants for their commitment in the projects. Without them, transdisciplinary research would not have been possible; everyone has therefore made a decisive contribution to the success of the project to date. Thanks also go to the German Federal Ministry of Education and Research (BMBF) and the Helmholtz Association for promoting our research and innovation. Finally, we thank Helga Kuechly for her contributions to an early version of the manuscript.

**Conflicts of Interest:** The authors declare no conflict of interest. The funders had no role in the design of the study; in the collection, analyses, or interpretation of data; in the writing of the manuscript, or in the decision to publish the results.

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
