# Peer review of "The Value of Citizen Participation in Technology Assessment, Responsible Research and Innovation, and Sustainable Development"

_sustainability, doi:10.3390/su132111613_

Round 1

Reviewer 1 Report

Research and innovation can be at the root of transformative changes in our society. To stimulate these changes to align with the needs and values of the society, a framework for responsible research and innovation (RRI) has been proposed. The authors aligned the three citizen participation projects (TeQfor1, TERRAIN and Nachtlicht-BÜHNE), pointing out similarities and differences according to the level of participation in different project phases. The authors found that the added value of citizen participation in Technology Assessment (TA), RRI, and Sustainable Development (SD), that is in the context of systemic and therefore complex societal transformation.

Author Response

Thank you for your positive and constructive feedback on our article. You will find the responses to the individual points of your review in the pdf file.

Reviewer 2 Report

The article is interesting and deals with issues related to the participation of citizens in scientific research. It is a less recognized problem in the literature on the subject. My general remark concerns the technical construction of the article. In my opinion, it is necessary to indicate what is the research gap of this paper. I also propose to complete the information about the added value of this paper. I suggest supplementing the information about the limitations and the direction of future research. 
In my opinion, the authors should also formulate research questions. Without all these elements, the article takes the form of a report rather than a scientific paper. 

Author Response

Thank you very much for your positive and constructive feedback on our article!! You will find the responses to the individual points of your review in the pdf file.

Reviewer 3 Report

The research is very interesting and it provides a useful contribution to the reflection about the variety of citizen's participation in research. However the manuscript remains to vague and to far from primary data.

Sentences such as "For example, citizens have confirmed to us that
they now have a better understanding of scientific processes through participation" (page 11) are too disconnected from what the auhors actually collected, and how they collected a data that supports such a claim.

Overall the 3 examples look very interesting and we would be very happy to ahev more factual information such as quotes from participants of these programs.

For example for TeQfor1 differences between adults and children who are mentionned to be users, would be very interesting to be exposed in the paper.

In table 1, line 2, column 2 "the final research questions were determined in a common effort". This is again too vague. This a very important step in such participatory processes, we need to know more how this happened, how intense was this common effort, etc.

Currently the paper requires major revisions as the readers lack primary, factual data to assess how well the claims made by the authors (which sound reasonable) are really supported.

I beleive the authors do have in hand all the information at stake, I would recommend that they provide more elements in the paper, maybe complementing with an attached supplementary data document.

Author Response

Thank you very much for your positive and the constructive feedback on our article!! You will find the responses to the individual points of your review in the pdf file.

Round 2

Reviewer 3 Report

The article can be read more easily and has been improved for some aspects.

However, I still feel that the reader lacks adequate proofs based on quotations, and more substantial data about what has been precisely observed in order to support the claims made by the authors.

I feel very unconfortable reading statements such as "according to participants’ statements during breaks and after the workshop", could we have any quotation? At least an example of these "statements"? And how reliable is a statement made during a break? Has it been recorded?

Author Response

Dear reviewer, thank you very much for the careful diligence regarding our revision and our data. You will find a response to your comment in the pdf file. With best regards

Round 3

Reviewer 3 Report

Dear authors, I think your three experiences are very interesting.

I leave it to the editing team to make a decision whether to publish your paper or not.

You have clarified your perspective, which is to say think as you see it from your stand-point of researchers, not of the one of citizens.

Still you make claims of "demonstration" that are inappropriate in the absence of data. See "The three case studies presented in the paper add to this, by demonstrating the importance of reducing the separation  of  science  and  society,  and  revealing  the  advantages  of  involving  citizens  in  research and innovation"

Either you change completely the standpoint of the paper, sharing your non-documented "feelings", "impressions" and then the paper is ok to me. It is a "perspective" or "opinion" paper which is highly valuable for the debate.

Then these are not proper "case studies". If they want to claim demonstrations and call these "case studies", this needs to be based on the kind of proper evaluation you call for, with collected data, selected quotations, etc.

I would consider that your paper need to be rejected if attempting to be a demonstration, but warmly accepted as an opinion paper.

Author Response

Dear reviewer, many thanks again for your comments! Please see the attached pdf file for our response. With kind regards, the author team

Round 4

Reviewer 3 Report

This last version is a significant improvement. It clearly reads as an opinion paper. Great.

Please just remove the 2 occurences of "demonstrate" and the 2 occurences of "confirm", and replace by more moderate assertions, in line with an opinion paper rather than a demonstration, proof or confirmation that would require data.

Author Response

Dear reviewer,

Many thanks for your positive feedback and the comment! We are pleased that the changes led to a further improvement of our manuscript. We have replaced the occurrences of “demonstrate” (three times) and “confirm” (only once as we could not find a second one).

With kind regards,

the authors
